

# Aquatic insect community structure revealed by eDNA metabarcoding derives indices for environmental assessment

Noriko Uchida[1,2], Kengo Kubota[2], Shunsuke Aita[3] and So Kazama[2]

[1] International Research Institute of Disaster Science, Tohoku University, Sendai, Miyagi, Japan
[2] Department of Civil and Environmental Engineering, Tohoku University, Sendai, Miyagi, Japan
[3] School of Engineering, Tohoku University, Sendai, Miyagi, Japan

## ABSTRACT

Environmental DNA (eDNA) analysis provides an efficient and objective approach for monitoring and assessing ecological status; however, studies on the eDNA of aquatic insects, such as Ephemeroptera, Plecoptera, and Trichoptera (EPT), are limited despite its potential as a useful indicator of river health. Here, we investigated the community structures of aquatic insects using eDNA and evaluated the applicability of eDNA data for calculating assessment indices. Field surveys were conducted to sample river water for eDNA at six locations from upstream to downstream of two rivers in Japan in July and November 2016. Simultaneously, aquatic insects were collected using the traditional Surber net survey method. The communities of aquatic insects were revealed using eDNA by targeting the cytochrome oxidase subunit I gene in mitochondrial DNA via metabarcoding analyses. As a result, the eDNA revealed 63 families and 75 genera of aquatic insects, which was double than that detected by the Surber net survey (especially for families in Diptera and Hemiptera). The seasonal differences of communities were distinguished by both the eDNA and Surber net survey data. Furthermore, the total nitrogen concentration, a surrogate of organic pollution, showed positive correlations with biotic environmental assessment indices (i.e., EPT index and Chironomidae index) calculated using eDNA at the genus-level resolution but the indices calculated using the Surber net survey data. Our results demonstrated that eDNA analysis with higher taxonomic resolution can provide as a more sensitive environmental assessment index than the traditional method that requires biotic samples.

## INTRODUCTION

Stream ecosystems are threatened by global climate changes and anthropogenic impacts, including damming, water abstraction and land-use changes (*WWF, 2016*). For sustainable development and resource use of freshwater, there is a need to effectively manage stream environments, which requires effective methods and indicators to measure and assess environmental impacts. Aquatic insects are commonly used as indicators of environmental health due to their high sensitivity to deterioration of water quality.

Corresponding author
Noriko Uchida,
noriko.uchida.d1@tohoku.ac.jp

They compose a core of the ecological food web in river ecosystems by feeding on producers and being preyed upon by higher consumers. Thus, monitoring of aquatic insect fauna is an effective method for assessing the environmental and ecological status. However, traditional survey methods such as kick net and Surber net sampling are subject to limitations. First, traditional field sampling processes result in data bias because the success and quality of a survey depend on the ability and skills of investigators and the accessibility of sampling sites. In addition, sampling methods that directly capture organisms inherently involve damage to natural habitat and organism. Second, the subsequent process of sorting and morphological identification is time-consuming and requires expertise in taxonomic identification (*Baird & Hajibabaei, 2012*; *Haase et al., 2006*). These limitations have been obstacles in performing high-frequency and long-term biological monitoring.

The use of environmental DNA (eDNA) is a novel biological monitoring method that can be used to overcome these limitations. Due to the simple sampling method involved (i.e., grab sampling of water, soil, etc.), eDNA monitoring can reduce the sampling bias caused by individual investigators (*Rees et al., 2014*; *Smart et al., 2015*). Moreover, it can minimize the bias associated with accessing the site because water and suspended materials, including eDNA, are transported and mixed through different environments. Therefore, eDNA can detect not only lotic animals but also lentic ones (*Fernández et al., 2018*; *Macher et al., 2018*; *Deiner et al., 2016*). This sampling method can also overcome ethical issues such as habitat disturbance and animal sacrifice associated with field sampling because it requires only nonbiotic samples. Furthermore, DNA-based identification can quickly provide results with higher taxonomic resolution than morphological identification (*Carew et al., 2013*; *Elbrecht & Leese, 2015*; *Hajibabaei et al., 2011*; *Serrana et al., 2018*). In addition, the necessary skills to analyze DNA can be acquired over a shorter time than those required for morphological identification. Although the characteristics of eDNA remain obscure (e.g., production and degradation rates and transportation dynamics), it can provide ecological information that is unobtainable via traditional survey methods. For example, eDNA can be used to evaluate the biodiversity across almost the entire fauna (fish fauna: *Ushio et al., 2018*) or across multiple phyla (phylum in eukaryotes: *Deiner et al., 2016*).

Aquatic insects such as Ephemeroptera, Plecoptera, Trichoptera (the abbreviated name of these groups in combination is EPT), and Diptera have been gradually targeted in eDNA studies in pursuit of effective river management (*Bista et al., 2017*; *Fernandes et al., 2018*; *Fernández et al., 2018*; *Hajibabaei et al., 2019a*; *Macher et al., 2018*; *Mächler et al., 2019*). However, there is a lack of information regarding the spatial and temporal differences of community structures of aquatic insects revealed by eDNA (*Bush et al., 2019*; *Roussel et al., 2015*). Furthermore, examining the possible use of eDNA data for assessing environmental status focusing on aquatic insects in stream ecosystems is required.

Therefore, the present study aimed to evaluate the applicability of eDNA in environmental assessments. First, we revealed the community structures of aquatic insects using eDNA and the traditional Surber net survey. Subsequently, we investigated whether eDNA data and the Surber net survey data can discern the spatial and temporal differences

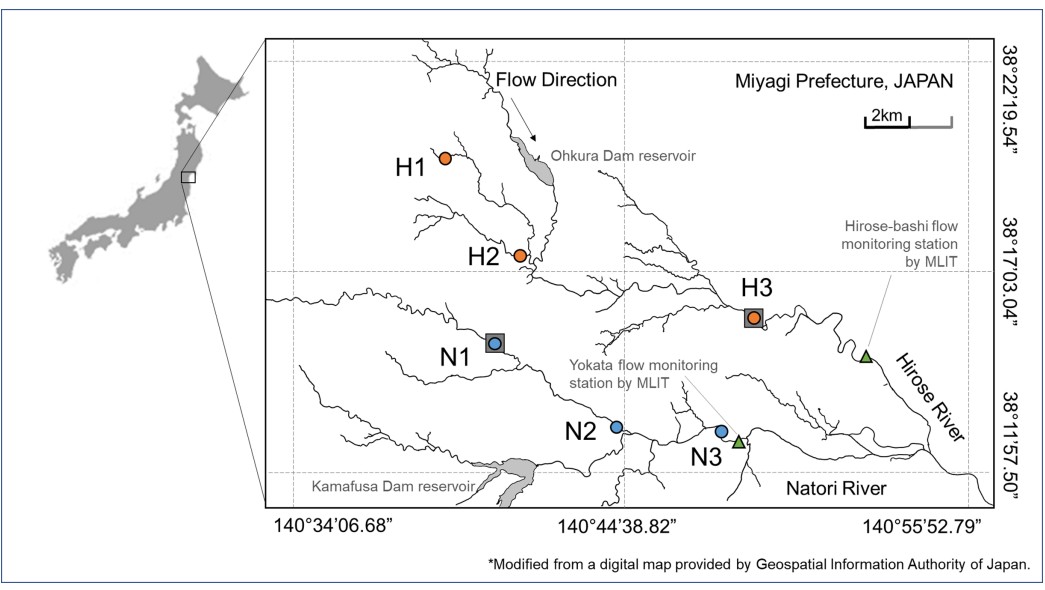

**Figure 1 Study area.** Sampling sites in Hirose River (from the upmost site, H1, H2, H3; shown as orange circles) and in Natori River (N1, N2, N3; shown as blue circles) in northeast Japan. This map was modified using a digital map provided by the Geospatial Information Authority of Japan.

of aquatic insect communities. We also evaluated the relationships between water quality parameters and assessment indices derived from each method.

## MATERIALS AND METHODS

### eDNA sampling, filtration, and DNA extraction

Field samplings were conducted at Hirose River and Natori River located at the Natori River basin, Miyagi Prefecture, northeast Japan. These are temperate rivers that originate in the mountains and flow through the hills at their middle reaches and through urbanized flatlands at their lower reaches and finally flow into the Pacific Ocean. The length of the channel of Hirose River is 45.2 km and its catchment area is 315.9 $km^2$. Natori River is 55.0 km long and its catchment area is 623.0 $km^2$ (not including the Hirose River Basin). Sampling was conducted in July and November 2016 at six locations from upland to lowland regions along the two rivers (sites H1–H3 and N1–N3; Fig. 1; Table S1).

Water samples for eDNA analysis were collected at the same sites and on the same days. The plastic bottles for eDNA sampling were sterilized with 10% chlorine bleach (Kao Corporation, Tokyo, Japan), rinsed with tap water in the laboratory, and washed thrice with river water at the collection site before sampling. At each site, flowing surface water was collected and transported to the laboratory on ice in a cool box. Water samples were filtered on the same day using vacuum filtration with 47-mm diameter glass fiber filters with a 0.7-μm pore size (GF/F, Whatman, 1 L/filter, referring to *Mächler et al. (2016)*). These filtered samples were stored at −20 °C until DNA extraction. DNA was extracted from the filters through lysis using proteinase K at 56 °C for 30 min. Then, the supernatant obtained was subjected to phenol–chloroform–isoamyl alcohol extraction and ethanol

precipitation. Eventually, the elution was purified using the OneStep PCR Inhibitor Removal Kit (Zymo Research, Irvine, CA, USA) with a final volume of 100 µl.

## Library preparation and sequencing

The cytochrome oxidase subunit I (COI) gene region in mitochondrial DNA was amplified from extracted DNA using the universal primer for invertebrates developed by *Folmer et al. (1994)*. The primer set of LCO1490 (5′-GGT CAA ATC ATA AAG ATA TTG G-3′) as the forward primer and HCO2198 (5′-TAA ACT TCA GGG TGA CCA AAA AAT CA-3′) as the reverse primer resulted in an amplification of a 658-bp fragment. For library preparation, a three-step polymerase chain reaction (PCR) was conducted. The first PCR was performed in a total volume of 20 µl PCR mixture comprising 10 µl of Taq$^{TM}$ HS Low DNA (TaKaRa, Kyoto, Japan), 0.4 µl each of 10 µM forward and reverse primers, 17.2 µl of ultrapure water and 2.0 µl of template DNA. The PCR conditions were as follows: 35 cycles at 94 °C for 5 s, 50 °C for 30 s, 68 °C for 10 s; and a final extension at 68 °C for 7 min. The fragment size of amplicons and the concentrations were verified by electrophoresis using the Agilent 2100 Bioanalyzer DNA7500 kit (Agilent, Santa Clara, CA, USA). PCR products were purified using the Agencourt AMPure XP (Beckman Coulter, Brea, CA, USA), and the purified products were used as templates for the following PCR. The second PCR was performed using Ex Taq Hot Start Version (TaKaRa, Kyoto, Japan) to add the overhang sequences that required amplification with the Nextera XT Index Kit for Illumina MiSeq analysis. The PCR conditions were as follows: 94 °C for 2 min; followed by 5 cycles of 94 °C for 30 s, 50 °C for 30 s, 72 °C for 30 s; and a final extension at 72 °C for 5 min. The amplicons were purified in the same manner as those obtained from the first PCR, and the purified products were used as templates for the next PCR. The third PCR was performed using Ex Taq Hot Start Version and Nextera XT Indice Kit v2 Set A (Illumina, San Diego, CA, USA). The PCR conditions were followed: 94 °C for 2 min; followed by 8 cycles of 94 °C for 30 s, 50 °C for 30 s, 72 °C for 30 s; and a final extension at 72 °C for 5 min. After purification, the final PCR amplicons were quantified using the Qubit dsDNA High Sensitivity Kit. The sequencing of prepared libraries was performed using MiSeq, according to the manufacturer's instructions.

## Bioinformatics

The sequence lengths of the amplicons were 658 bp so the forward and reverse reads in our study could not be merged when using the MiSeq Reagent Kit v3 (600 cycles). *Elbrecht & Leese (2017)* have demonstrated that invertebrate species could be identified at the reverse side of the COI region through an in silico PCR approach. Therefore, we conducted a subsequent analysis using the reverse side sequence. Initially, the raw sequence reads were subjected to the Trimmomatic v0.36 software to discard low-quality sequences and read sequence lengths of <150 bp. The filtered reads were clustered into operational taxonomic units (OTUs) using QIIME (*Caporaso et al., 2010*), with an identity cutoff value of 97%, which is a common approach for invertebrate metabarcoding analyses (*Macher et al., 2018*). Subsequently, OTUs with singleton sequences were removed.

The most frequently occurring sequences in each OTU were extracted as representative sequences. The assignment was performed against 3,433,026 sequences retrieved from the National Center for Biotechnology Information (NCBI) database using the following search criteria: cytochrome (all fields) AND oxidase (all fields) AND mitochondrion (filter). After the assignment, eight orders (i.e., Ephemeroptera, Plecoptera, Trichoptera, Diptera, Coleoptera, Odonata, Megaloptera and Hemiptera) that mostly include aquatic insect species were extracted using the QIIME script "filter_taxonomy_from_table.py." Subsequently, representative sequences of the extracted OTUs were subjected to a chimera check. Taxonomic identification was performed using a BLAST search and the QIIME script "assign_taxonomy.py." Because the traditional environmental assessment indices (%EPT, %Diptera and %Chironomidae) require at least family-level taxonomic identification, we employed two thresholds for taxonomic identification: 97% identity for genus-level assignation and 85% identity for family-level assignation. For macroinvertebrates, a threshold of 97% is commonly used for species/genus-level assignment (*Hebert, Ratnasingham & DeWaard, 2003*) but may cause a loss of sequence depth. According to our investigation and using a subset of sequence data of EPT registered in NCBI, the sequence identity for the COI region of EPT was 99% at the intraspecific level, 85% at the intragenus level, 83% at the intrafamily level and 80% at the intra-order level (see Text S1). Therefore, a BLAST assignment was conducted with a minimum identity of 97% (assigned at the genus-level) and 85% (assigned at least at the family-level) and a maximum e-value of $10^{-50}$ (*Fernández et al., 2018*). To compare the communities among samples, we subsampled the number of sequences in each sample by a uniform number. According to the smallest numbers of sequence reads in the samples, either 250 reads for the family-level analysis or 150 reads for the genus-level analysis were randomly selected (see Table S2).

## Aquatic insect sampling using a Surber net survey and measurement of environmental parameters

After eDNA sampling but within the same day, traditional aquatic insect collection was conducted using the Surber net survey method. A Surber net of 250-μm mesh size in a $30 \times 30$-cm quadrat at randomly selected riffle and pool habitats at each site (total collection area: 0.18 m$^2$/reach) was used. Collected invertebrates were placed in 99.5% ethanol and morphologically identified using a stereomicroscope (Leica MZ APO; Leica, Germany) by referring to the identification key for the aquatic insects of Japan (*Kawai & Tanida, 2018*). Because morphological identification was difficult for some aquatic insects, particularly Chironomidae and some Baetidae individuals, population abundance and richness were summarized at the family level.

At the same time as aquatic insect sampling, environmental parameters such as water temperature, electrical conductivity (EC; TOA-DKK CM-21P; Japan), and pH (TOA-DKK HM-20P; Japan) were measured in the field. River water samples of 50 ml were collected from each site to obtain the concentrations of total phosphorus (TP) and total nitrogen (TN) measured in our laboratory using a QuAAtro-2HR (BLTEC Corporation, Japan).

## Community structure analysis

We assessed the dissimilarity in community structures of aquatic insects using eDNA or Surber net survey data. Community dissimilarities were calculated based on the Sørensen index (binary Bray–Curtis index) using the presence/absence of OTU data for eDNA and the detected taxa for the Surber net survey using the "vegan" package (*Oksanen et al., 2019*) in R ver. 3.4.0 (*R core team, 2018*). Using the ordination of dissimilarity, nonmetric multidimensional scaling (nMDS) was performed to visualize the similarity in community structures and the "metaMDS" function in the "vegan" package. Furthermore, the correlations between community structures and environmental parameters (i.e., water temperature, EC, TN and TP) were determined using the "envfit" function in the "vegan" package.

## Environmental assessment indices

The applicability of biological information obtained from eDNA to existing environmental assessment indicators, namely, EPT index, Diptera index and Chironomidae index (*Reynoldson & Metcalfe-Smith, 1992*), was evaluated. These indices are the ratios of the number of individuals/richness of EPT taxa to the total number of individuals/richness of the eight orders observed in the samples (Ephemeroptera, Plecoptera, Trichoptera, Diptera, Coleoptera, Odonata, Megaloptera and Hemiptera). For the Surber net survey data, the EPT index was calculated using the abundance or the richness at the family/genus-level of EPT. The Diptera/Chironomidae index uses the same method as the EPT index, but using Diptera/Chironomidae instead of EPT. For eDNA-analyzed samples, each index was calculated using the richness of OTUs or groups by taxonomic name at the assigned family/genus-level. Here, the OTU richness refers to the number of OTUs included in the sample, and the assigned family/genus richness refers to the number of families/genera included in the sample (see Text S2 for formulas).

# RESULTS AND DISCUSSION

## The community structure of aquatic insects revealed by eDNA analysis

A total of 1,235,176 sequences (50,728–168,413 sequences/sample) passed the sequence quality filter (Table S2 for the detail of metabarcoding outputs). These sequences were used to create OTUs based on 97% sequence identity. As a result, 90,948 OTUs were formed. Among these, 66,176 OTUs comprised just one sequence (singletons), which were excluded from the analysis. Finally, a total of 1,169,000 sequences (47,443–161,461 sequences/sample) generating 24,773 OTUs were analyzed. After a BLAST search, we found that 8.0% of the total sequences were assigned to aquatic insects at the family level (sequence identity ≥85%, see "Materials and Methods") and only 4.1% of them were assigned at the genus level (sequence identity ≥97%) (Table S2).

eDNA metabarcoding detected 93 families and 104 genera before subsampling, and after subsampling, 63 families and 75 genera were detected (sequence depths were 250 reads for the family-level assignment and 150 reads for the genus-level assignment; see Tables S2 and S3 for the OTU tables before subsampling). A total of 26 families were common with the Surber net survey results. Even after subsampling, the total number

of taxa detected by the eDNA method was nearly double than that detected by the Surber net survey method (Table S4). Specifically, eDNA detected 27 genera of Chironomidae (Diptera); however, these genera could not be distinguished by morphological identification. In addition, eDNA detected taxa that were mostly distributed in riparian/terrestrial habitats (e.g., Culicidae; Diptera, Cicadidae; Hemiptera) and lentic habitats (Aeshnidae and Epiophlebiidae; Odonata).

According to the subsampling results of eDNA, the taxa detected in both the months were 26 families/25 genera, those detected in July alone were 26 families/41 genera, and those detected in November alone were 11 families/9 genera. Among these, we found that the three families (Ephemerellidae, Chironomidae and Simuliidae) were commonly detected among all sites in both the months (see Table S5 for details of assignment results at the family level). Conversely, a number of unique taxa were detected in the communities at site H1 (11 genera in July and 9 genera in November; see Table S6 for details of assignment results at the genus level).

Previous studies have reported that compared with the use of a traditional survey method, the use of eDNA in lotic systems tends to enable the detection of more taxa (*Macher et al., 2018*); this is in contrast with the case in pond systems wherein DNA transportation is very low (*Hajibabaei et al., 2019a*). This is because DNA is transported downstream in lotic systems, which results in the additional detection of upstream communities that are overlooked by traditional methods. It has been reported that fish eDNA is decomposed and transported after release from organisms, with a 73% decrease in eDNA concentration within 900 m downstream of the source (*Nukazawa, Hamasuna & Suzuki, 2018*). Even 50–250 m downstream of the source, eDNA was not reported to be detected when the target organisms' abundance or biomass is small (*Jane et al., 2015*; *Pilliod et al., 2014*). Thus, the DNA sampled in rivers probably includes some DNA originating from abundant organisms, within an approximate distance of 1 km upstream. While the source materials of eDNA differ depending on the organism (e.g., mucus for fish (*Takeuchi et al., 2019*), saliva for terrestrial mammals (*Rodgers & Mock, 2015*; *Ushio et al., 2017*), and exuvia for aquatic arthropods (*Deiner & Altermatt, 2014*)), the nature of eDNA in lotic systems may exist in a similar manner. The interval between sampling sites in our study was approximately 3–5 km; therefore, the eDNA contamination among samples was assumed to be negligible. In addition, *Jo et al. (2017)* have shown that amplification of longer DNA fragments (719 bp vs. 127 bp) is more effective in reflecting real time biological information. The length of the amplicons in the present study was relatively long (658 bp); therefore, the community structures obtained using eDNA were mainly based on DNA that might have been generated recently and transported from a closer area. In addition, eDNA can be used to detect taxa that are usually difficult to capture using the Surber net survey method in lotic systems such as terrestrial organisms (*Deiner et al., 2016*; *Mächler et al., 2014*). As shown in previous studies, terrestrial and semiterrestrial taxa were also detected in our samples. These results indicate that the eDNA sampled from river ecosystems provides a diverse taxonomic list that differs from the traditional Surber net sampling method.

From the results of the family-level assignment, nine families were not detected in any eDNA samples but were detected with Surber net sampling, namely, two ephemeropteran (Isonychiidae and Siphlonuridae), one plecopteran (Chloroperilidae), one trichopteran (Apataniidae), one dipteran (Blephariceridae), and four coleopteran (Gyrinidae, Hydrophilidae, Psephenidae and Ptilodactylidae) taxa. Of these, Isonychiidae, Blephariceridae, Hydrophilidae, Psephenidae and Ptilodactylidae have mismatched sequences with the primers used in this study, which results in failure of PCR primer amplification. Therefore, the primers should be modified or new primers should be developed to analyze these five families. Some refined primer sets for the metabarcoding of aquatic invertebrates have been developed (*Elbrecht & Leese, 2017*; *Hajibabaei et al., 2012*). *Hajibabaei et al. (2019b)* also suggested that the use of multiple universal primers enables coverage of a broader range of taxa.

In addition to the primer issue, the sequence identity threshold used for taxonomic identification can be another problem for eDNA analysis. To evaluate the discrepancy between the reference library and the query sequence, we investigated intraspecific, intrageneric and intrafamilial genetic identity (Text S1). As a result, an 85% identity threshold at the family level and a 97% identity threshold at the genus level were employed for taxonomic assignment in this study. However, this threshold might not have been achieved by some species and thus gone undetected. To overcome this issue, reference sequence data should be accumulated. Geographically separated intraspecies have low sequence identity of the COI gene (*Takenaka & Tojo, 2019*). Therefore, the accumulation of genetic information of local aquatic insects and the construction of a database are necessary to improve the taxonomic assignment and to avoid false negative results. There is an urgent need to overcome these issues because failure to detect some specific taxonomic groups could directly affect the assessment results based on the richness of taxa.

## Relationships between communities and environmental parameters

Community dissimilarities among all samples using the Sørensen index for eDNA analysis and Chao index for Surber net survey data were plotted on nMDS axes (Fig. 2). Visually, nMDS showed monthly differences among communities (July or November). Ordination was significantly correlated with water temperature in all three datasets (function "envfit"; eDNA at the genus level: $R^2 = 0.56$, $p = 0.016$, eDNA at the family level: $R^2 = 0.81$, $p = 0.001$ and Surber net survey data at the family level: $R^2 = 0.57$, $p = 0.021$; Table S7). *Bista et al. (2017)* have shown that eDNA targeting the chironomid community (Diptera) in a lake system can distinguish seasonal differences across communities. Similarly, the present study demonstrated that eDNA analysis targeting the aquatic insect community in a river system revealed seasonal differences.

In addition, nMDS showed that the uppermost site of the Hirose River (H1) was plotted in isolation from the other sites in both months and for all three datasets (Fig. 2). This was understandable because the landscape of H1 differs from those of the other sites; it is a mountain stream, and its inhabitants differ from those found in other middle and lower reaches. Indeed, eDNA analysis detected multiple unique taxa at H1 that made differences among communities (Tables S5 and S6). Conversely, in the Surber net survey, the

 

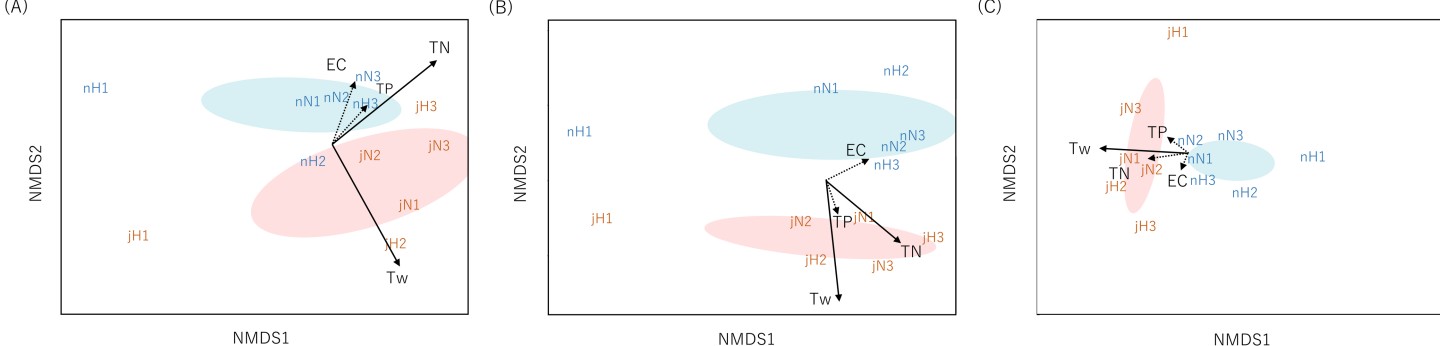

**Figure 2 Non-metric multidimensional scaling (NMDS) using Sørensen dissimilarity index (P/A data).** Each panel shows the communities derived from (A) eDNA at genus-level identification, (B) eDNA at family-level identification, (C) Surber net (at family-level identification). eDNA data were based on OTU richness with subsampled by 150 reads depth for genus level and 250 reads depth for family level. The same month is enclosed by ellipses (orange, July; blue, November). Site name and month (j, July; n, November) are displayed. The environmental parameters are shown by arrows (solid: $R^2 > 0.5$ with $p$-value < 0.05, dotted: $R^2 \leq 0.5$ with $p$-value $\geq$ 0.05). The length of the arrow is proportional to the correlation between parameters and the community ordination.

differences among communities were revealed by the absence of taxa that could be found in the other sites, rather than the presence of unique taxa at H1 (Table S4). Thus, both eDNA analysis and Surber net survey clearly illustrated differences among communities depending on landscapes (mountain stream or middle/lower reach), whereas the factors attributable to the differences among communities differed between methods.

Furthermore, the ordination based on eDNA data was significantly correlated with TN concentration (function "envfit"; eDNA at the genus level: $R^2 = 0.50$, $p = 0.043$, eDNA at the family level: $R^2 = 0.51$, $p = 0.046$); however, no such correlation was noted for the Surber net survey data. TN concentration may not only be a direct proxy for organic pollution, but also an indirect proxy for site characteristics. That is because agricultural and urban land use increase in the lower area in our study area, so the larger the stream order, the higher the TN concentration (Table S1). Previous studies reported that eDNA can distinguish geographical changes in communities in river systems for various fauna (fungi: *Matsuoka et al., 2019*; macroinvertebrates: *Hajibabaei et al., 2019b*; *Fernández et al., 2019*). Similarly, the present study showed that the community differences revealed by eDNA analysis were related to the site characteristics by water quality rather than by geographical location.

## Environmental assessment indices derived from eDNA

The relationships between the TN concentration and biological environmental assessment indices (%EPT, %Diptera and %Chironomidae) were examined (Fig. 3) because TN could be assumed as a chemical indicator of water pollution. The results showed that %EPT and %Chironomidae derived from eDNA at genus-level resolution showed sufficient effect sizes with significant rank correlation with TN (Spearman's rank correlation; %EPT: $r = -0.59$, $p = 0.049$, %Chironomidae; $r = 0.69$, $p = 0.014$; Table S7). However, the eDNA and Surber net survey results at family-level resolution did not show significant correlations. These results indicate that when eDNA data are obtained at a higher taxonomic resolution, the sensitivity of biological indices to environmental factors

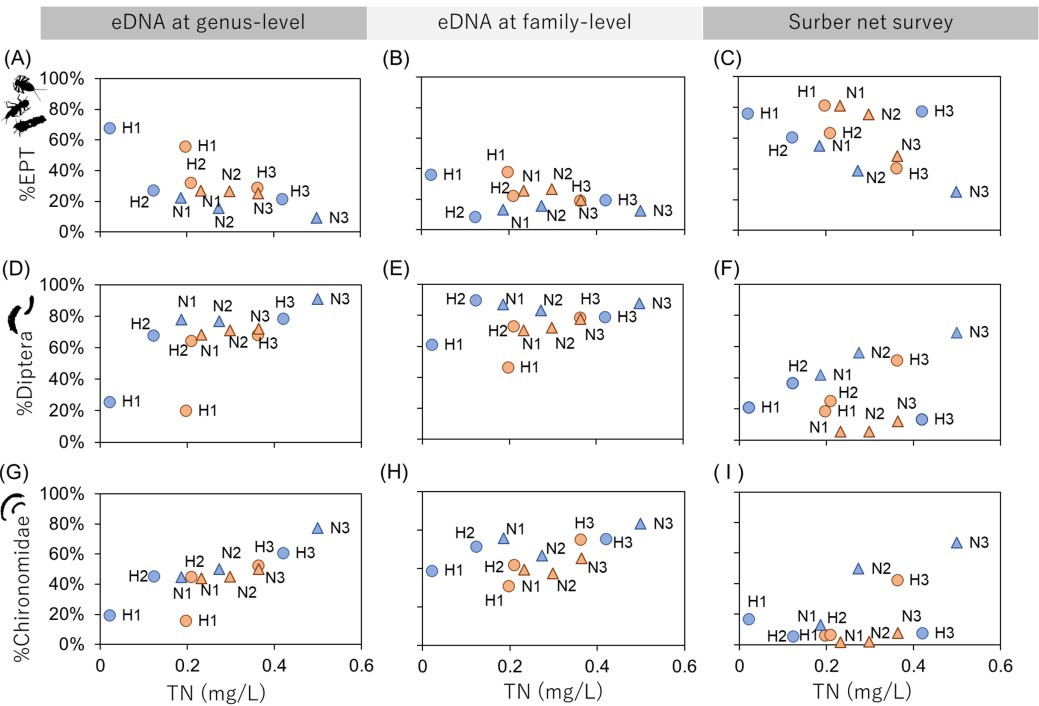

**Figure 3 Relationships between biological assessment indices and TN concentration.** The first (A–C), second (D–F), and third (G–I) rows show %EPT, %Diptera and %Chironomidae, respectively. The first, second and third columns show results based on eDNA at genus-level identification, eDNA at family-level identification and Surber net survey, respectively. The indices are calculated using OTU richness data for eDNA (subsampled) and abundance data for Surber net data. Seasonal differences are represented by colors (orange, July; blue, November) and river differences are represented by plot styles (circle, Hirose River; triangle, Natori River), respectively.

can be improved. Conversely, the sensitivity could be impaired if biological indices are obtained using coarse taxonomic resolution.

*Emilson et al. (2017)* have reported that assessment indices (i.e., the EPT index and Chironomidae index) derived from DNA metabarcoding using macroinvertebrate tissue samples were highly correlated with indices derived from the morphological survey. The present study demonstrated that the indices obtained from eDNA can also be used as a new assessment method. Although the biological indices obtained from eDNA vs. the traditional method were only compared in terms of TN concentration in the present study, new environmental indicators can be developed using eDNA data by comparison with more chemical pollution indicators such as biochemical oxygen demand and chemical oxygen demand.

While our manuscript was under review, a study that detected macroinvertebrate eDNA and applied this to the environmental status assessment of a river was reported (*Fernández et al., 2019*). In that report, the environmental assessment score from the IBMWP index was calculated based on the presence/absence of indicator macroinvertebrates at family-level identification. The report demonstrated that eDNA data could be used for the monitoring program that they used. Conversely, the present study showed that the EPT

and Chinoromidae indices calculated using OTU richness required genus-level resolution and showed a clearer response to organic pollution compared with family-level resolution.

## CONCLUSIONS

eDNA can be used to describe differences among community structures of aquatic insects in two seasons in river systems. In addition, compared with the ordination of community derived from traditional sampling methods, that derived from eDNA analysis was correlated with the degree of water pollution. EPT and Chironomidae indices at the genus level derived from eDNA analysis data showed significant correlations with TN concentration, whereas indices derived from Surber net survey and eDNA analysis data at the family level did not. Environmental assessment indices based on ecological information but not requiring biotic samples have significant advantages such as they can be applied to places where the capture of organisms is restricted. In addition, eDNA analysis can derive benefits related to sampling such as minimal sampling effort, high taxonomic resolution, and high applicability to a broad range of species. We believe that eDNA analysis is useful for monitoring the long-term trends of changes in ecological community structure associated with environmental changes such as climate change and other anthropogenic activities, and it facilitates environmental assessment with nonbiotic samples.

## ACKNOWLEDGEMENTS

The authors would like to thank Mr. Kengo Watanabe (Tohoku University, Japan) and Dr. Yasuyuki Takemura (National Institute for Environmental Studies, Japan) for their assistance on field survey and experiments. We also greatly appreciate Dr. Kei Nukazawa (University of Miyazaki, Japan) and Dr. Yasuhiro Takemon (Kyoto University, Japan) for providing meaningful comments on our paper.

### Funding

This study was supported by the Japan Society for the Promotion of Science (JSPS) (grant numbers: 16H02363) and JSPS Research Fellowship (grant number: 17J02158) and through the Program for Leading Graduate Schools, "Inter-Graduate School Doctoral Degree Program on Global Safety." There was no additional external funding received for this study. The funders had no role in study design, data collection and analysis, decision to publish, or preparation of the manuscript.

### Grant Disclosures

The following grant information was disclosed by the authors:
Japan Society for the Promotion of Science (JSPS): 16H02363.
JSPS Research Fellowship: 17J02158.
Program for Leading Graduate Schools.
## Competing Interests

The authors declare that they have no competing interests.

## Author Contributions

- Noriko Uchida conceived and designed the experiments, performed the experiments, analyzed the data, prepared figures and/or tables, authored or reviewed drafts of the paper, and approved the final draft.
- Kengo Kubota conceived and designed the experiments, analyzed the data, authored or reviewed drafts of the paper, and approved the final draft.
- Shunsuke Aita performed the experiments, authored or reviewed drafts of the paper, and approved the final draft.
- So Kazama conceived and designed the experiments, authored or reviewed drafts of the paper, and approved the final draft.

## DNA Deposition

The following information was supplied regarding the deposition of DNA sequences:

The nucleotide sequence data is available in the DDBJ sequence Read Archive (DRA): DRA008293 and at BioProject PRJDB8220.

## Data Availability

The raw sequence data are available in Figshare: Uchida, Noriko; Kubota, Kengo; Aita, Shunsuke; Kazama, So (2020): Aquatic insect community structure revealed by eDNA metabarcoding derives indices for environmental assessment. figshare. Dataset. DOI 10.6084/m9.figshare.8082188.v1.

Additional raw data is available in the Supplemental Files.

## Supplemental Information

Supplemental information for this article can be found online at http://dx.doi.org/10.7717/peerj.9176#supplemental-information.

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
