# Peer review of "Aquatic insect community structure revealed by eDNA metabarcoding derives indices for environmental assessment"

_PeerJ, doi:10.7717/peerj.9176_

## Round 0.1 · original submission · Major Revisions

Both reviewers are in accord that your paper has merit, but that significant revision of the paper is required, and certainly this will require re-analysis of your data. Moreover, as this is not current practice in metabarcoding studies, you must provide clear evidence of the validity of using sequence read numbers as a proxy for abundance of sequenced organisms, since there is little evidence in the published literature that this is in fact possible.

Reviewer 1 ·

Basic reporting

The manuscript is generally clear and easy to follow. Figures and tables are appropriate for the material

Experimental design

The experimental design is sound

Validity of the findings

I have some concerns with the validity of the findings given specific elements of the analysis. I have described these in more detail in the general comments below.

Additional comments

I am pleased to provide this review of Manuscript #37104, “Aquatic insect community structure revealed by eDNA metabarcoding derives indices for environmental assessment” by Uchida et al. The manuscript describes the use of eDNA for bioassessment of aquatic insect community structure. Although the experimental methods are fundamentally sound, I believe the conclusions are suspect based on several limitations of the analysis. Specifically, assumptions about number of reads being a good surrogate for relative abundance, reliance on family level analysis, and use of a reference library approach vs. an OTU or ASV approach may mask differences between the traditional and molecular approaches and lead to erroneous conclusions. Therefore, I recommend that the article be returned for major revisions before being considered for publication in Peer J.

The assertion that read counts indicate relative abundance (Line 63) is not a consensus opinion in the literature. Issues such as amplification bias during PCR and size specific variation in DNA shedding rates make the relationship between relative abundance and read count suspect (Cristescu and Hebert 2018). Although studies in closed systems, such as aquaria or mesocosms have shown relationships between abundance and read counts, these relationships do not necessarily hold true in natural environments, particularly in streams where environmental factors such as temperature, flow, and organic matter content can affect degradation and transport differently. The authors should focus on bioassessment measures that do not rely on abundance measures in order to more accurately compare traditional and molecular approaches.

The authors decision to cluster OTUs at 85% similarity leads to artificially coarse grouping which increases the likelihood of similarity between methods but may not be meaningful from an assessment perspective. Similarly, the higher number of families detected using eDNA based analysis is likely reflective of higher taxonomic resolution (i.e. the clusters likely represent genus or species vs. family). I suggest that the authors repeat the analysis but cluster at 97%, which would be comparable to species level analysis. Even if the OTU clusters cannot be named, it would allow a more direct/appropriate comparison of the two approaches in terms of relative performance for assessment.

The reliance on taxa in the reference library also presents a limitation in the analysis. As the authors state, 84% of OTUs were classified as “no blast hit” or “no assignment” (Line 209), which limits the actual comparison between the two methods. DNA based analysis is increasingly relying on use of unnamed OTUs or amplicon sequence variants (ASVs) as the basis of new biological indices. I suggest that the authors not limit the analysis to taxa that can be matched to libraries and instead compare basic richness measures as well.

Additional minor comments

In the presentation of results on the ability of the methods to discern site and spatial differences, it would be helpful to have the authors clearly state the hypothesis about site differences – are they expected to be different based on landscape position or stressors? – I would be easier to interpret findings in the context of clearly stated expectations.

Line 270 – are the PCA plots provided either in the text or supplemental material? If not, they should be.

Figure 3 does not really illustrate differences by spatial position – please clarify in the text.

What is known about site H1 that makes in an outlier on the NMDS plots? Some explanation for its clear difference from the other sites would be helpful. We can often learn a lot by looking at conditions at outlier sites.

At the very end of the Discussion (Line347), the authors correctly point out that comparisons between taxa and OTU richness may be confounded by the fact that eDNA at the sites could include OTUs that washed down from upstream. This is a critically important consideration that warrants much more discussion.

Literature cited
Cristescu, ME and PDN Hebert. 2018. Uses and Misuses of Environmental DNA in Biodiversity Science and Conservation. Annual Review of Ecology, Evolution, and Systematics. 49:209–30

Reviewer 2 ·

Basic reporting

See attached PDF for detailed comments.

Experimental design

See attached PDF for detailed comments.

Validity of the findings

See attached PDF for detailed comments.

Additional comments

See attached PDF for detailed comments.

Annotated reviews are not available for download in order to protect the identity of reviewers who chose to remain anonymous.

---

## Round 0.2 · Major Revisions

Both reviewers have indicated that you have failed to seriously address the many significant concerns they had previously identified in your paper. I was inclined to reject your paper at this point, but I have decided to give you a second chance to address the major shortcomings indicated by the two reviewers (both of whom had previously reviewed your original submission). I believe that your paper has merit, and that it could yet see publication in PeerJ if their concerns are fully addressed. I will be checking your revision closely.

Reviewer 1 ·

Basic reporting

see comments to editor

Experimental design

see comments to editor

Validity of the findings

see comments to editor

Additional comments

see comments to editor

Reviewer 2 ·

Basic reporting

See below and in attached PDF.

Experimental design

See below and in attached PDF.

Validity of the findings

See below and in attached PDF.

Additional comments

I would like to thank the authors for taking the time to fully address some of the comments made in the original review. First, while the writing style has improved for the original manuscript text, this does not extend to the sections where comments were addressed in response to the reviewers and these should be reviewed. Second, unfortunately, the goal of this paper still focuses on attempting to make the DNA metabarcoding data conform to the traditional data and the calculation of traditional metrics. For example, the insistence of using directly comparable data between the two methods, which has penalized the DNA metabarcoding data (e.g. only using family-level data, focusing the analyses on abundance data inferred from read counts). Also note that the plot between the Shannon diversity metrics, calculated for both methods, is not convincing. At a minimum, the two axes should be aligned - and a 1:1 line added to the plot. The two methods are just different ways of observing the river system – both have advantages and disadvantages in their sampling, in their data properties, and in their potential to be used for monitoring. Also, different metrics can be used in observing and quantifying change, and we don’t have to continue to use the same metrics with new data with its new properties (although I understand that legacy of data for long-term monitoring is important but that shouldn't preclude looking at new ways to observe and monitor systems).

I would argue that a better approach would be to treat the two data sets maximizing their properties rather than forcing the data to conform e.g. use abundance for the Surber net data but acknowledging the coarser taxonomic resolution and then using presence-absence data for the DNA metabarcoding (particularly in the light of the observed bias) at a higher resolution (probably genus level). A small section of the paper could explore the use of abundance inferred from read counts as this is interesting – but, as you acknowledge, potentially limited. The focus could still be on the spatial and temporal differences among the sites and systems – and ordinations can be compared using the different data sets provided by the two methods allowing you to say whether similar patterns are observed, despite the different data properties. These are just suggestions, and I defer to the Editor to make the final decision.

Annotated reviews are not available for download in order to protect the identity of reviewers who chose to remain anonymous.

---

## Round 0.3 · Minor Revisions

Please fix the issue identified by Reviewer 2 regarding nMDS axis labels.

Reviewer 1 ·

Basic reporting

no comment

Experimental design

no comment

Validity of the findings

no comment

Additional comments

I have reviewed the revised manuscript by Uchida et al. “Aquatic insect community structure revealed by eDNA metabarcoding derives indices for environmental assessment” (Manuscript #37104). The manuscript has been significantly revised and now addresses my substantive comments on the original version. I appreciate the authors addressing my concerns over surrogate measures of abundance, level of taxonomic resolution and use of the reference library. Although, I still think it would have been interesting to explore a taxonomy-free analysis with this data set, I believe the current version of the manuscript is acceptable for publication. My only remaining comment pertains to Section 3.2, where the authors should acknowledge that some of the spatial differences attributed to temperature may also be associated with flow differences (which were not measured as part of this study).

Thank you for the opportunity to review the revised manuscript. I recommend that this significantly improved version be accepted for publication.

Reviewer 2 ·

Basic reporting

The paper is much improved but I still find the writing stilted at times and difficult to follow. There are still a number of grammatical/spelling errors throughout the manuscript that need to be addressed prior to publication.

Experimental design

N/A

Validity of the findings

N/A

Additional comments

I would like to thank the authors for addressing the comments raised in the previous review. The paper is much improved (although see comment on text/language above including caption descriptions). The authors need to be careful in their nMDS plot which currently shows the axes. As you cannot use nMDS axis scores in any quantifiable interpretation as they are dimensionless then it is usually not recommended to plot the axis coordinates (but I leave this to the Editor for the final decision).

---

## Round 0.4 · accepted · Accept

Thanks for making these final revisions - your paper is now acceptable and I am recommending that the paper be accepted without the need for further revision or review.